# Bioactivity of an Experimental Dental Implant with Anodized Surface

**DOI:** 10.3390/jfb12020039

**Published:** 2021-06-07

**Authors:** Maria Fernanda Lima Villaça-Carvalho, Juliani Caroline Ribeiro de Araújo, Juliana Mariano Beraldo, Renata Falchete do Prado, Mari Eli Leonelli de Moraes, Luiz Roberto Coutinho Manhães Junior, Eduardo Norberto Codaro, Heloisa Andrea Acciari, João Paulo Barros Machado, Natal Nerímio Regone, Anderson Oliveira Lobo, Fernanda Roberta Marciano, Luana Marotta Reis de Vasconcellos

**Affiliations:** 1Department of Bioscience and Oral Diagnosis, Institute of Science and Technology, UNESP—São Paulo State University São José dos Campos, São Paulo 12245-000, Brazil; mfervillaca@hotmail.com (M.F.L.V.-C.); julianicraraujo@hotmail.com (J.C.R.d.A.); juliana.beraldo@unesp.br (J.M.B.); eli@ict.unesp.br (M.E.L.d.M.); manhaes@ict.unesp.br (L.R.C.M.J.); 2Department of Chemistry and Energy, School of Engineering, Guaratinguetá Campus, UNESP—São Paulo State University, Guaratinguetá, São Paulo 12516-410, Brazil; eduardo.codaro@unesp.br (E.N.C.); heloisa.acciari@feg.unesp.br (H.A.A.); 3Associated Laboratory of Sensors and Materials, National Institute for Space Research, INPE, São José dos Campos, São Paulo 12227-010, Brazil; machadopaulo@gmail.com; 4Department of Aeronautic and Communication Engineering São João da Boa Vista Campus, UNESP—São Paulo State University, São João da Boa Vista, São Paulo 13876-750, Brazil; natalregone@sjbv.unesp.br; 5LIMAV—Interdisciplinary Laboratory for Advanced Materials, BioMatLab, UFPI—Federal University of Piaui, Teresina 64049-550, Brazil; lobo@ufpi.edu.br; 6Department of Physics, UFPI—Federal University of Piauí, Teresina 64049-550, Brazil; marciano@ufpi.edu.br

**Keywords:** anodizing, microtomography, nanotechnology, osseointegration

## Abstract

Background: Several studies proved that anodic oxidation improves osseointegration. This study aimed to optimize osseointegration through anodization in dental implants, obtaining anatase phase and controlled nanotopography. Methods: The division of the groups with 60 titanium implants was: control (CG); sandblasted (SG); anodized (AG): anodized pulsed current (duty cycle 30%, 30 V, 0.2 A and 1000 Hz). Before surgery, surface characterization was performed using Atomic Force Microscopy (AFM), Scanning Electron Microscopy (SEM), X-ray Dispersive Energy Spectroscopy (EDS) and Raman Spectroscopy. For in vivo tests, 10 New Zealand white rabbits received an implant from each group. The sacrifice period was 2 and 6 weeks (*n* = 5) and the specimens were subjected to computed microtomography (μCT) and reverse torque test. Results: AFM and SEM demonstrated a particular nanotopography on the surface in AG; the anatase phase was proved by Raman spectroscopy. In the μCT and in the reverse torque test, the AG group presented better results than the other groups. Conclusion: The chemical composition and structure of the TiO_2_ film were positively affected by the anodizing technique, intensifying the biological characteristics in osseointegration.

## 1. Background

The anodizing process has received considerable attention, as it is an efficient and low-cost reproducibility technique, in addition to exhibiting suitable surface modification for cellular activities, improving surface properties through nanotopography [1,2]. These changes in the surface of the implants can accelerate the bone repair process, as well as increase bone deposition [3,4], playing an important role in the osseointegration process, being the interface between cell-substrate crucial for the success of the biomaterial [5]. The nano-dimensioned resources can simulate the cellular environment [6] and favor the proliferation and adhesion of mesenchymal and osteoblastic cells due to the increase in the surface area of the biomaterial [1,7].

In addition, the anodizing process modifies the amorphous oxide film in a layer of crystalline oxide on the implant surface [8]. Current techniques that obtained a layer of crystalline and uniform oxide, showed improvements in the results in this layer when compared to that formed naturally, in the atmosphere. Thus, the performance of the TiO_2_ implant as a biomaterial may be associated with the conversion of its amorphous form into crystalline [9]. The crystalline forms of TiO_2_ are rutile, used in cosmetics and paints, with thermodynamic stability as a characteristic; bronquita, intermediate stage of crystallinity and anatase, which is the crystalline phase, manufactured at lower temperatures and has biocompatibility [10].

Metals showed greater resistance to corrosion and abrasion and also showed improvement in osseointegration [11]. The hydrophilicity of the crystalline surface (anatase phase) compared to the amorphous one promoted cell scattering in the film when the interaction of osteoblasts in commercially pure titanium (cpTi) and Ti alloy was analyzed showing amorphous and crystalline titanium dioxide [7].

Several studies showed that anodic oxidation has improved osseointegration [12]. However, little research studied anodic oxidation on the surface of commercial implants [8]. The present work had the purpose of evaluating the nanotopography and chemical structure of the crystal line in the anatase phase obtained by a new route in the anodization process and in the newly formed bone and implant fixation.

## 2. Materials and Methods

### 2.1. Anodic Oxidation and Implants

In this study, 60 Ti external hexagon implants, grade IV, rounded conic alapex and self-drilling screw were used, measuring 8.5 mm × 3.75 mm in diameter (Titanium Fix Company, Sao Paulo, Brazil). The implants were randomly divided into three experimental groups of 20 implants each: control (CG-machined surface); sandblasted (SG-machined surface which was blasted with aluminum oxide followed by subtraction by nitric acid); anodized (AG-machined surface subjected to the anodization technique with application of pulsed current, 0,2 A, 30 V and 1000 Hz, for 4 h).

The group sandblasted was commercially acquired from Titanium Fix^®^. To obtain the group anodized, the surface treatment was performed in Chemistry Laboratory at São João da Boa Vista Campus of São Paulo State University (SP, Brazil). Briefly, the machined implants (screws) were properly fixed on a titanium plate. The anodization of the implants occurred after cleaning the surface. For the anodizing process, the titanium plate and a copper plate were used as anode and cathode, respectively. Both plates were placed into a beaker containing 1.0 mol/L H_2_SO_4_ solution as electrolyte. The parameters used for the anodization were: 0.2A, resulting current, 30 V of applied potential, and 1000 Hz of frequency pulses, for 4 h, at room temperature. Despite many anodization procedures having been described in the literature presenting shorter times, the samples of the present research presented the best surface properties after 4 h (data not shown). Electrical parameters were monitored using a digital oscilloscope, model MO2061, Minipabrand; a pulsating square wave rectifier, GI21P-10/30 model, of the company General Inverter. The implants were cleaned using ultrasound followed by sterilization with 25kGy at Embrarad (Sterilization Unit LTDA, Cotia, SP, Brazil).

To characterize morphologically the implant surface, Scanning Electron Microscopy (PhilipsXL-30 FEG, PHILIPS, Leuven, Belgium) and Atomic Force Microscopy (AFM) Multimode Nanoscope V (Veeco Instruments Inc. New York, NY, USA) were used. To analyze any superficial damage and changes in the chemical composition of the implant surfaces, the surface analysis was performed before and after surgery by reverse torque.

To determine the chemical composition of the anodic film, Raman spectroscopy (Horiba, Kyoto, Japan) and X-ray dispersive energy spectroscopy (EDS, Horiba, Kyoto, Japan) were used. The technique of electrochemical impedance spectroscopy (EIS) was used to determine the physicochemical characteristics and the corrosion resistance of the film of the experimental surfaces (sand blasting and anodizing). To find the bands positioned in the titanium anatase region, a Raman Horiba Scientific T64000 spectrometer (Horiba, Kyoto, Japan) was used, with this TiO_2_ phase being more crystalline and biocompatible. In the analysis by EDS, an energy-dispersive detector and the Bruker Esprit 1.9 software was used for the chemical microanalysis. The concentration of chloride ion found in blood plasma is similar to 0.9%, so in the EIS measurements were performed in 0.9% NaCl. Autolabpotentiostat/galvanostat (PGSTAT302N model, Eco. Chemie BV, Utrecht, Netherlands) was used. Briefly, a conventional cell showing three electrodes and each sample represented the working electrode. A Pt electrode was used as auxiliary and all potentials were recorded against a saturated Ag/AgCl electrode. EIS diagrams were registered at OCP by applying a 10 mV sinusoidal potential through a frequency domain of 100 kHz to 10 mHz.

### 2.2. Surgical Procedures

First, the protocols were approved by the Ethics in Research Committee of the Institute of Science and Technology of São José dos Campos from the State University of São Paulo–UNESP (02/2014-PA/CEP) and ARRIVE was respected.

New Zealand white male rabbits (*n* = 10) 5 months old, on average, weighing about 4 kg were used, which were kept in individual cages and fed standard solid rations and water ad libitum at the vivarium of the Science and Technology Institute of Sao Jose dos Campos-ICT, UNESP.

The present study allows eliminating the interference between individuals, since one implant of each experimental group was placed in right tibia of each rabbit. In addition, different destructive tests were performed, due to the placement of an implant of each group also in left tibia. Five animals were randomly chosen for each evaluation period, in order to describe similarly aspects of the healing process.

After weighing the animals, general intramuscular anesthesia was performed with a mixture of 13 mg/kg of xylazine hydrochloride (Anasedan-Vetbrands) and 33 mg/kg of ketamine (Dopalen^®^-Agibrands do Brasil Ltd.a). The local anesthesia used was a 3% prilocaine hydrochloride compound associated with 0.03 IUU/mL felipressin (Citanest3% ^®^-Dentsply). The left and right tibiae were subjected to scraping and antisepsis with iodinated solution [4] followed by an incision made in the proximal third of the tibia in the region medial. Six implants were installed in each tibia, one from each group. After the implant placement surgery, the layers were sutured with silk thread no. 4 (Ethicon^®^/J&J Medical Devices) and antisepsis, with the use of iodized alcohol. Postoperative antibiotics were used (6,000,000 IU benzatinabenzyl penicillin, procaine benzyl penicillin, benzyl penicillin potassium and dihydroestrostptomycin sulfate based on dihydroestrostptomycin sulfate) (Pentabiotic-FortDodge), intramuscularly at the dose of 1, 35 mL/kg in the immediate postoperative period (48 h) and ketoprofen analgesic (Ketofen, Fort Dodge Animal Health, Fort Dodge), 1 mg/kg subcutaneously every 24 h for 3 days. In the post-surgical period, the animals were kept in individual cages with food and water ad libitum. Euthanasia in the period of 2 and 6 weeks (*n* = 5) was performed with deep general anesthesia (propofol 10 mg/kg) intravenously followed by administration of an intravenous potassium chloride vial.

The right side of the specimens removed after euthanasia were stored in a buffered formaldehyde solution to be subjected to Computed Microtomography (μCT). While the specimens on the left side, which were removed, were stored in a Ringer at −20 °C, to be kept in similar conditions to the body to be subjected to the torque removal test.

### 2.3. Removal Torque Testing

After euthanasia, the left side tibial specimens were removed and immobilized around. The reverse torque test, using a digital torque meter (Mark-10 Corporation, New York, NY, USA) was performed. Counterclockwise rotation was applied and the maximum torque values (N.cm) required for bone fracture at the bone-implant interface were measured.

### 2.4. Computed Microtomography (μCT)

Computerized microtomography (μCT) was performed with 360-degree scan rotation of the parts, using monochrome X-rays, with 89 kV and 275 µA, 0.1 copper filter (SkyScan 1176, Skyscan, Kontich, Belgium) on the fragments of right tibia with the implant. The analysis was performed after obtaining the images using the NRecon software (version 1.6.6.0) for image reconstruction; data Viewer, CTAnalyzer (BrukerMicroCT) to evaluate the parameters of bone volume (VB), trabecular thickness (Tb.Th) and the relationship between bone volume and trabecular volume (BV/TV) by selecting a rectangular region (ROI) creating a volume of interest (VOI) and the creation of the 3D image of each analyzed fragment, adjusting a histogram to differentiate the newly formed bone tissue according to density using the CT-Vol software (v.1.14.4, Kontich, Belgium).

### 2.5. Cytotoxicity Evaluation by MTT

Through sequential enzymatic digestion, osteogenic cells were isolated from male newborn rats aged two to four days (Rattus norvegicus, Albinus, Wistar variant) (*n* = 30). Subsequently, they were sown in the implants and cultivated as described by Andrade et al. (2015) [13]. The medium was changed every three days, and the evolution of the cells was evaluated under an inverted microscope. After 3 days and 10 days, the MTT solution of 3-(4,5-dimethylthiazol-2-yl) -2,5-diphenyltetrazole (Sigma) was inserted to the wells with cells and kept at 37 °C for 4 h [13]. After this period of incubation, the supernatant was removed and the samples were washed with PBS, then 1.0 mL of isopropanol (0.04 mol/L HCl in isopropanol) was added to each well. The analysis was performed using with an EL808IU spectrophotometer (Biotek Instruments, Winooski, VT, USA) at 570 nm. The data were expressed as absorbance. The control used for this test was the well containing only cells.

### 2.6. Statistical Analysis

All statistical tests were performed with the aid of the software GraphPad Prism (version 6.0, GrahPad, San Diego, SA, USA) by means of non-parametric test Kruskal Wallis was used to determine the significant differences, followed by a Test Dunn post-hoc test. The level of significance adopted was 0.05.

## 3. Results

### 3.1. Implants Characterization

SEM analysis demonstrated topographical differences among implants surfaces. CG presents the smooth appearance of (Figure 1A). SG presented a topography characteristic of subtraction process, which formed irregular valleys, with different depths and sizes (Figure 1B). Finally, the AG presented nanotexturized surface with topography more uniform and valleys less depth (Figure 1C,D).

When analyzing the images obtained by the AFM, the presence of traces inherent to the machining stage of dental implants, a textured surface in the micrometer range and the adequate uniformity of a nanotextured surface, a characteristic pertinent to the anodizing process, was verified.

The verification of chemical elements, performed by means of Dispersive Energy Spectroscopy (EDS) of an implant in each group, demonstrated the predominant presence of titanium in the CG In the SG, Ti and Al were identified; Ti and O were presented at the AG. In addition to Ti and O, some other components were also found in the AG implant film.

In the Raman spectra (Figure 2A), the characteristic frequency bands of the anatase were observed in the anodized groups (AG). In AG, bands positioned in the anatase regions were observed (Figure 2B). The frequency bands were identified as: 147 (±2.8) cm^−1^, 392.8 (±4.3) cm^−1^, 515.2 (±5.3) cm^−1^, 513.14 (±4.7) cm^−1^, 628.8 (±10.2) cm^−1^. No peak of anatase was observed in the sandblasting group (SG) (Figure 2A).

Figure 2C show the EIS spectra obtained from the complex planar format. Extrapolating the capacitive semicircle to the point of intersection with the real axis can estimate the polarization resistance R_p_, which is determined in the frequency spectrum generated by the low frequency region. For the commercial samples processed by blasting, the smallest diameter of the capacitor arc was obtained. The R_p_ value is about 100 kΩ, which is about 10 times the R_p_ value of anodized samples; this shows that the Sandblasted Group has a greater susceptibility to corrosion than the Anodized Group; presenting the greatest resistance to corrosion extrapolating the Y axis of the graph.

### 3.2. Removal Torque Testing

The data obtained in the removal torque tests were subjected to descriptive analysis for each group (Figure 3). The highest values were observed in anodized implants compared to the other groups at 2 weeks. 

A statistical difference was observed only in the 2-week periods, in which the CG differed statistically from the AG group (*p* < 0.05). There was no statistical difference between groups in relation to osseointegration, in the period of 6 weeks. (*p* = 0.0714).

### 3.3. Micro-Computed Tomography

The data obtained through the descriptive analysis and the inferential statistical results of the groups are represented in Figure 4 and Figure 5, in both periods. The mean values of AG implants were higher when compared to the other groups, with the exception of the parameter Tb.N (trabecular number) in 2 weeks. The comparison of bone volume using Kruska Wallis showed no statistically significant difference in the period of 2 weeks (*p* = 0.1139) and 6 weeks (*p* = 0.3271). A similar result was observed in parameter Tb.N, in which the period of 2 weeks *p* = 0.6298 and 6 weeks *p* = 0.8151. With regard to BV/TV, the AG showed greater results, differing statistically when compared to the CG and SG at 2 weeks (*p* = 0.0012). In none of the parameters or periods, statistical differences between CG and AG could be observed. However, in the 6-week period, there was no significant differences between groups (*p* = 0.6298) when analyzed BV/TV. In the 3D images obtained from the microtomography, it was possible to observe that there was a greater amount of cortical bone for AG implants, both in the period of 2 weeks to 6 weeks; compared to other implants (Figure 6).

### 3.4. Cytotoxicity Evaluation by MTT Assay

Within three days, the groups showed no statistical difference between groups and between the control group (well with cells only). While the SG presented a higher percentage of viable cells in the period of 10 days, this differed statistically with the control group (*p* < 0.05). Most of the groups analyzed exhibited greater cell viability over time and none of the groups showed cytotoxicity to the cells. The results can be seen in Figure 7.

## 4. Discussion

The crystalline structure found in the anatase phase of TiO_2_ leads to an appropriate anatomy for mineral growth on the surface of the biomaterial [14]. This occurs because the mineralization in the formation of bone tissue is promoted by hydroxyapatite crystals, which are nucleated in this phase [15,16].

Among the techniques used to modify the surface of the implants [17,18,19] the anodization process consists of forming a nanotopography that favors cellular activity and has the ability to interact with fluid and bone tissue. As an electrochemical method, of low cost and reproducibility, it also presented positive results in relation to bioactivity [2,20].

The titanium anodizing technique in orthopedic implants and in the dental field has been used. An example of the latter is TiUnite^®^, Nobel Biocare AB, Gothenburg, Sweden. Thus, it is evident that the morphological characteristic of TiO_2_ is a preponderant aspect in the studies, as well as what was observed in a study by Mu-Hyon Kim et al. (2015) [1] in which the authors evaluated implants subjected to the anodization process and observed the presence of a rough surface on a nanometric scale. In another study, by Pinheiro et al. (2014) [21], anodization was obtained in 1.0% Na_2_SO_4_ solution, applying 100 V for 1 min and the authors observed a nanotextured surface. El-Wassefy et al. (2014) [22], used a solution of H_2_SO_4_, 200 V for 4 min, with subsequent treatment for 1 h at 600 °C and obtained similar results.

Based on previous studies [21,22], in this study we used a time for the anodizing process of 4 h, which is longer and more innovative in the relevant literature, in order to form a nanoscale roughness in TiO_2_ and a more chemical composition biocompatible with the interest of obtaining the anatase phase of TiO_2_ without the need for heat treatment at high temperatures. For the process, in addition to the time already mentioned, other parameters were used (1.0% H_2_SO_4_ solution as electrolyte, 0.2 A, 30 V and 1000 Hz) resulting in the morphological changes observed in the images obtained by SEM and AFM. While the anatase phase was observed in AG, as evidenced by Raman spectroscopy, using the bands observed in positions in the anatase region [23]. In addition, the anodized implants showed an increase in the thickness of TiO_2_ without generating residues through the anodization process, presenting a different result from that of Pinheiro et al. (2014) [21], where it was possible to observe the presence of residues in the electrolyte solution.

Impedance measurements reveal the working electrodes in a passive state with varying degrees of film compaction. In the Bode format, the wide peaks recorded for the phase angle at intermediate frequencies are very close to 90 degrees for all conditions evaluated and the slopes of Log (|Z|/Ω) vs. Log (f/Hz) are close to-1 A large peak can be indicative of the interaction of at least two time constants, related to two simultaneous interfacial processes [24]. These variations are in line with the previous study [25] and were attributed to an ideally capacitive behavior.

Among the various techniques for quantifying neoformed bone in bone defects [26], osseointegration assessed by μCT is convenient, since it does not cause damage to the sample and allows complete visualization of the bone, being a good tool to assess bone structure at around the implant [3], considered similar to conventional histomorphometry. Some studies in the literature performed μCT osseointegration histomorphometry. There are different methods and parameters to quantify the bone using μCT [3,27]. In this study, the analyses were performed in cross-sections, as well as the study by Vandeweghe et al. (2013) [3], data were collected along the axis of the implant at the beginning of the cortical bone. Based on the results obtained from the analyzed parameters (BV, BV/TV and Tb.Th), in this study we show that bone repair and BV/TV and Tb.Th osseointegration of the implants were accelerated in the groups that underwent the process anodizing in both periods. Bone repair in animals starts in the first week after the defect is made and ends in about 3–4 weeks. The use of studies with different periods of euthanasia [22], as well as this study, allows the comparison of the influence of biomaterials surfaces at the beginning of tissue repair with the stability of the implant in relation to its bioactivity. As demonstrated by Badrand El-Hadary (2007) [28] after longer periods of osseointegration, bone repair reaches a balance, where it is observed that histologically the tissue and vascular architecture remain constant.

The verification of bone fixation to the implant was obtained through the reverse torque test [22]. As in this study, changes in the properties of the oxide present on the surface of titanium implants have demonstrated significant positive effects on bone-implant fixation over a 3-week period [22].

Here, we use pre-osteoblastic cells from the skull of newborn rats to avoid misunderstandings in differentiating bone marrow stem cells in culture. It is an important experimental model for studies of bone metabolism, compatibility of biomaterials and bone integration [13]. Li et al. (2014) [29], after comparing the cell viability by MTT of commercially pure Ti samples submitted to anodization, demonstrated that a positive effect on bone cell differentiation and proliferation was attributed to the high roughness and hydrophilicity of the anodized surface. The increase in the percentage of viable cells in the longer evaluated periods observed in this study, was like that reported in previous studies [29,30]. It was also possible to observe that none of the implant groups showed toxicity to the cells, in both monitoring periods.

## 5. Conclusions

In this study, we evaluated the performance of nanotopography of titanium implants subjected to the anodizing process in bone repair in the osseointegration process. The results obtained from the sample characterization tests showed that the anodized group showed changes in the chemical and structural composition of the titanium oxide film, favorable to osteogenic activity. It was also possible to observe, through the micrographs obtained in the SEM, a nanotextured surface in this experimental group, which provided greater bone formation. None of the observed groups showed in vitro cell cytotoxicity. The AG group showed improvement in mechanical properties and bone repair around implants evaluated in vivo, in which there was an improvement in bone volume when compared to CG and SG group. From these results it is suggested that the process of anodizing the surfaces is beneficial and can be used in dental practice.

## Figures and Tables

**Figure 1 jfb-12-00039-f001:**
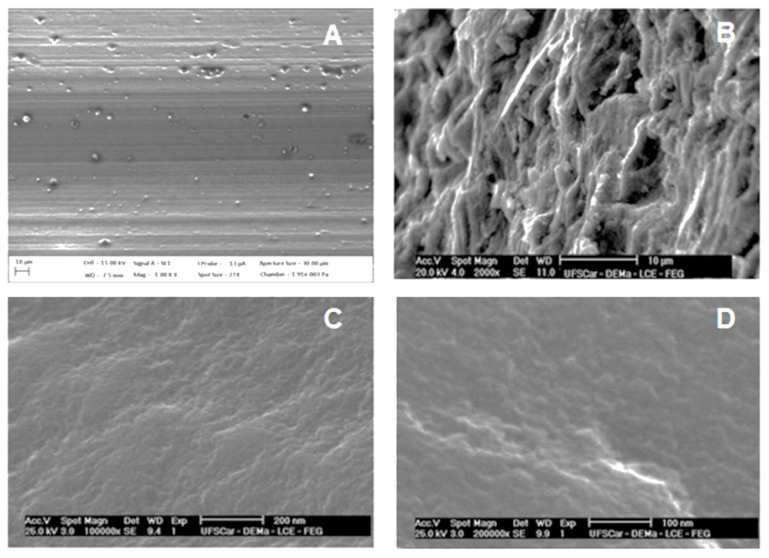
Images obtained from SEM. (**A**) CG showing grooves on the surface due to the machining process with original magnification of 1000× (10 μm); (**B**) SG showing some cuts due to the impact of the oxide particles on the surface, increasing the roughness by 2000 times (10 μm); (**C**) AG on the surface of the nanostructured implant increased the roughness by 100,000 times (200 nm); (**D**) The AG on the surface of the nanostructured implant increased by 200,000 times (100 nm).

**Figure 2 jfb-12-00039-f002:**
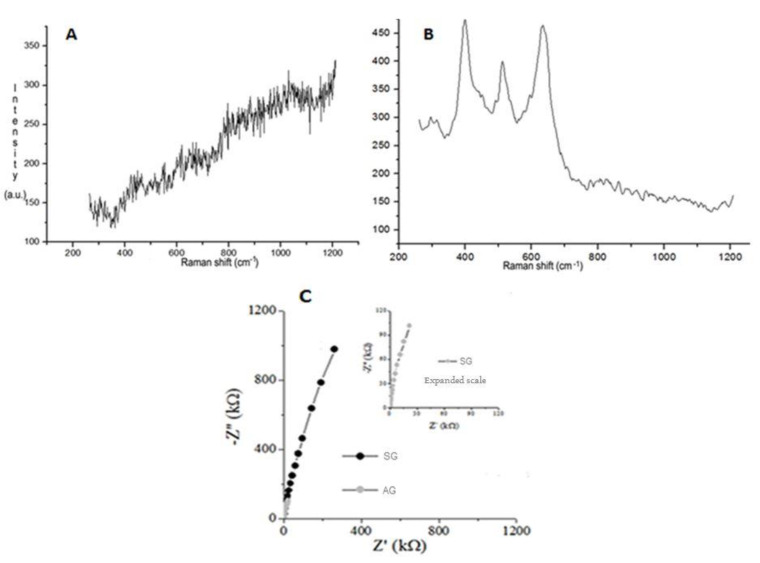
Chemical composition of surface implant (**A**) Raman spectroscopy of SG implant demonstrating the absence of anatase phase. (**B**) Raman spectroscopy of AG, by anodizing pulsed current, demonstrating the presence of anatase phase. (**C**) EIS spectra obtained in the form of the complex plane, the modified titanium surfaces in NaCl 0.9%. The curve of AG shows highest values than SG, then presents increase corrosion resistance.

**Figure 3 jfb-12-00039-f003:**
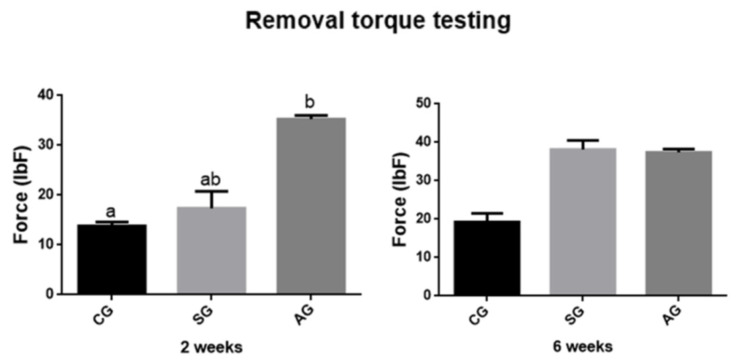
Torque test data. Effects of surface topography on osseointegration measured by reverse torque test. Anodized group presented higher values than control group at 2 weeks. Statistical differences are shown using different letters. (Kruskal Wallis and Test Dunn’s).

**Figure 4 jfb-12-00039-f004:**
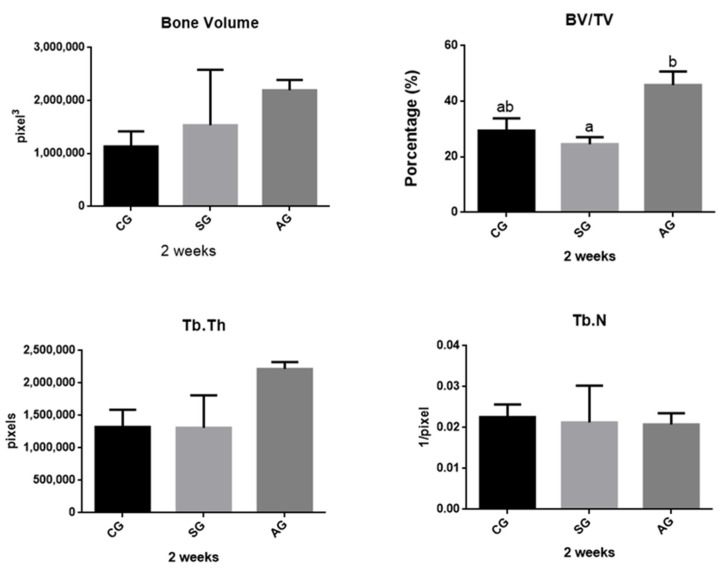
Micro-Computed Tomography data. BV, Tb. N and Tb. Th did not present differences between groups. The BV/TV rate showed a statistical difference between the AG and SG groups at 2 weeks. Statistical differences are shown using different letters. (Kruskal Wallis and Test Dunn’s).

**Figure 5 jfb-12-00039-f005:**
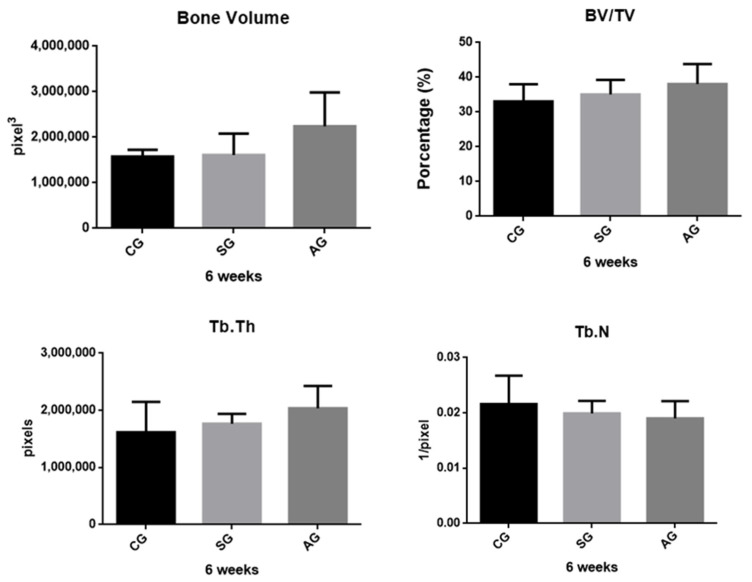
Micro-Computed Tomography data. BV, BV/TV, Tb. N, Tb. Th did not present differences between groups at 6 weeks (Kruskal Wallis and Test Dunn’s).

**Figure 6 jfb-12-00039-f006:**
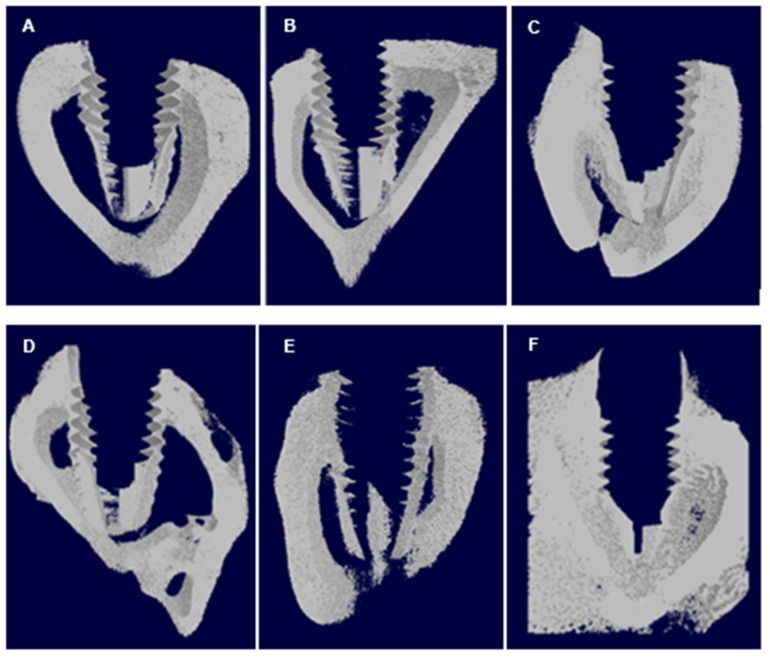
3D images obtained in the CT-Vol software. (**A**) GC, (**B**) SG, (**C**) GA within 2 weeks; (**D**) CG, (**E**) SG, (**F**) AG within 6 weeks. The central area is the region of the implant and the periphery is the selected bone tissue.

**Figure 7 jfb-12-00039-f007:**
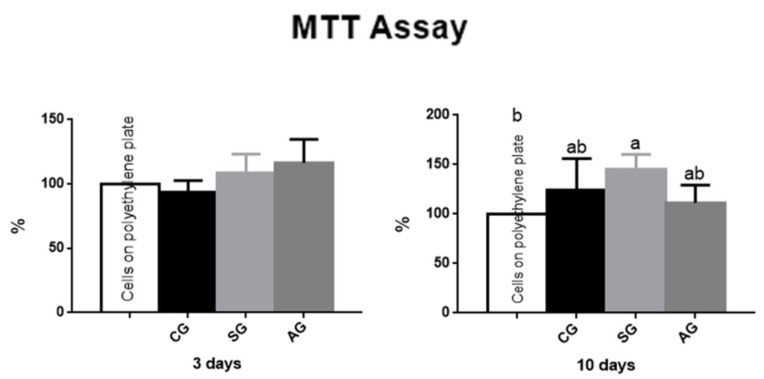
Data from MTT Assay. At 3 and 10 days, cytotoxicity was similar in the three experimental groups. Differences were only observed between control group (cells plated on polystyrene plate) and sandblasted group at 10 days. Statistical differences are shown using different letters. (Kruskal Wallis and Test Dunn’s).

## Data Availability

The datasets used and/or analyzed during the current study are available from the corresponding author on reasonable request.

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
