# Peer review of "Bioactivity of an Experimental Dental Implant with Anodized Surface"

_jfb, 2021, doi:10.3390/jfb12020039_

Round 1
Reviewer 1 Report
many sentences need to be rephrased to avoid plagiarism....
Author Response
Thanks for your consideration. Changes were made to the manuscript to avoid plagiarism. However, there are still words identified in plagiarism software, but they are unique words in the text, such as names of institutions, cities, prepositions and units of measures, used in the methodology.
Reviewer 2 Report
The article looks correctly formatted, the abbreviations are always explained the first time they appear and then only the abbreviation is used. Although short the introduction is focused on the topic and provides all the relevant information to adequately set the background for the sections of the articles to come. Materials and methods are described in details,. Statistics look correct and detailed. Results are clearly presented. Discussion and conclusion are supported by results but in the conclusion section I would very much appreciate if the authors could expand conclusion section
Author Response
Thanks for your consideration. We agreed with the suggestion and made the change to the conclusion.
Reviewer 3 Report
Dear authors !
I was given the opportunity to review Your experimental study investigating a novel design of dental implant surfaces.
The study design and execution is sound, the manuscript well written and well documented by figures and tables, the results discussed clearly and comprehensible.
Maybe it would be of high interest to the readers to present some additional figures of microCT and SEM-images to illustrate the differences between all groups regarding surface structure and osseointegration.
Thank You
Author Response
Thanks for your consideration. MicroCT figures and SEM images were added to the manuscript as proposed.
Reviewer 4 Report
Thanks for submitting this manuscript.
I have carefully read your manuscript with great interest.
It could be supporting the novelty of study by acquired data reliability on statistical analysis.
Major comments:
Authors must be clear statistical point (parametric or non-parametric test, normality, one-way ANOVA etc).
If there were only 5 experiments on Fig 2, 3, 4, there is not enough experimental number to apply parametric tests, as the normal distribution cannot properly be verified with only 5 experiments in each group. Therefore, the Shapiro-Wilk method (or any other test to verify the normality, for that matter) is useless, as the normal distribution cannot properly be verified. I strongly recommend that some data were used by Kruskal–Wallis test. Specially, pairwise comparisons were performed through the Mann-Whitney test.
Moreover, authors need to provide the representative images for each group in figure 3.
Authors will be able to re-analyze the results by appropriate statistical analysis, after that, it may happen that the results of some statistical comparisons may change and consequently the discussion.
Minor comments;
Line 29: … in the reverse touch test (miss-typo?)
Line 181: Control implant (GC) (miss-typo?)
Line 211: In Fig 1D bode format, need to explain for 4 line and symbols.
Line 218: what is CA group?
Line 221: in figure 2, need to indicate the 2- and 4- weeks period.
In figure 2, 3, and 4: what means the small alphabet (a, b, ab)? Need to clear the meaning.
Author Response
Major comments:
Authors must be clear statistical point (parametric or non-parametric test, normality, one-way ANOVA etc).
- If there were only 5 experiments on Fig 2, 3, 4, there is not enough experimental number to apply parametric tests, as the normal distribution cannot properly be verified with only 5 experiments in each group. Therefore, the Shapiro-Wilk method (or any other test to verify the normality, for that matter) is useless, as the normal distribution cannot properly be verified. I strongly recommend that some data were used by Kruskal–Wallis test. Specially, pairwise comparisons were performed through the Mann-Whitney test.
Moreover, authors need to provide the representative images for each group in figure 3.
Authors will be able to re-analyze the results by appropriate statistical analysis, after that, it may happen that the results of some statistical comparisons may change and consequently the discussion.
Answer: Thank you for your attention in reading the work. The statistics and graphs have been changed according to the proposed statistics.
Minor comments;
Line 29: … in the reverse touch test (miss-typo?)
Answer:Thanks for your consideration. It has already been corrected in the text.
Line 181: Control implant (GC) (miss-typo?)-
Answer:Thanks for your consideration. It has already been corrected in the text.
Line 211: In Fig 1D bode format, need to explain for 4 line and symbols.
Answer: The graphic representation was removed from the text, as it was similar to the graphic representation of the test showed in image 2C.
Both being representations of the same characterization test.
Line 218: what is CA group?
Answer:Thanks for your consideration. It has already been corrected in the text.
Line 221: in figure 2, need to indicate the 2- and 4- weeks period.
Answer: Thanks for your consideration. The change in Figure 2 has already been made.
In figure 2, 3, and 4: what means the small alphabet (a, b, ab)? Need to clear the meaning.
Answer: Thanks for your consideration. Information that different letters indicate statistical differences has been added to the image caption.
Round 2
Reviewer 1 Report
good job
Reviewer 4 Report
Well revised.